# Associations between academic achievement and weight status in a multi-ethnic sample of New Caledonian adolescents

Stéphane Frayon[1]*, Viren Swami[2,3], Guillaume Wattelez[1], Akila Nedjar-Guerre[1], Olivier Galy[1]

1 Interdisciplinary Laboratory for Research in Education, EA 7483, School of Education, University of New Caledonia, Nouméa, New Caledonia, 2 School of Psychology, Sport, and Sensory Sciences, Anglia Ruskin University, Cambridge, United Kingdom, 3 Centre for Psychological Medicine, Perdana University, Kuala Lumpur, Malaysia

* stephanefrayon@hotmail.com

**Data Availability Statement:** All relevant data are within the paper and its Supporting Information files.

## Abstract

Several studies have reported a negative association between obesity and academic achievement in school-aged children. In the Pacific region, the prevalence of adolescent overweight is high, but no study has considered issues of academic achievement in this population. To rectify this, we examined relationships between academic achievement and weight status in a multi-ethnic (European and Kanak) sample of New Caledonian adolescents. Objective anthropometric measures (height, weight, waist circumference) were obtained from European and Kanak New Caledonian adolescents ($N = 526$) between July 2018 and April 2019. Body mass index (BMI) and waist-to-height ratios (WHtR) were used as proxies of weight status. Ethnicity was self-reported and additional sociodemographic data (socioeconomic status, gender identity, urbanicity of residence, school remoteness) were extracted from relevant databases. Academic achievement scores were obtained from ninth grade national test in language, mathematics, history-geography, and sciences. Bivariate correlations showed that the associations between anthropometric indicators of weight status and academic achievement were significant in adolescents of European, but not Kanak, origin. Underweight and normal-weight European adolescents had significantly higher academic achievement than Kanak adolescents at the same weight categories. Additionally, BMI-$z$ was significantly associated with academic achievement after controlling for socio-demographic variables, but only in adolescents of European origin. Weight status appears to be associated with academic achievement in New Caledonia, but only in adolescents of European origin. Ethno-cultural understandings and experiences may shape the ways in which weight status affects academic achievement in this context.

**Funding:** This project was funded by the University of New Caledonia and the Fondation Nestlé France.

**Competing interests:** The authors have declared that no competing interests exist.

## Introduction

Escalating rates of overweight and obesity among children and adolescents represent a serious public health challenge for societies and healthcare systems worldwide [1, 2]. Comparisons of anthropometric data further suggest that the Pacific region has some of the highest rates of overweight in children and adolescents, with obesity prevalence at over 30% in some countries [3–7]. In fact, the ten countries with the highest prevalence of overweight in the world are all located in the Pacific region [8]. This is a major public health concern because overweight and obesity are associated with several non-communicable diseases (NCDs), such as hypertension, cancer, diabetes, and cardiovascular events [9–11]. Additionally, obesity and overweight are also associated with poorer psychological well-being, including symptoms of depression [12], disordered eating behaviours [13], body dissatisfaction [14, 15], and low self-esteem [16–18].

In addition to these associations with physical and mental health outcomes, overweight and obesity may also be associated with school-aged students' academic achievement. Thus, several reviews of the literature have reported a negative association between obesity and academic performance in school-aged children [19–22]. However, this association is more equivocal after controlling for covariates, including socioeconomic status, parental education, and/or physical activity [21], and a recent meta-analysis concluded that the association between body mass index (BMI) and academic achievement was weak ($r$ = -.11) [23]. Importantly, although this meta-analysis also found that the relationship was moderated by geographic region (with the association being stronger in Europe than in North America and Asia), it is notable that no previous study has focused specifically on populations from the Pacific. This is especially important because the limited evidence suggests that the association between weight status and academic achievement may be negligible in some nearby populations [24], although reasons for this are not fully understood.

One possible explanation is that academic achievement is not shaped by weight status *per se*, but is in fact mediated or moderated by related factors. Indeed, several mechanistic pathways through which overweight and obesity could affect academic achievement have been proposed, including school absenteeism linked to health issues [25], poor sleep quality [26], low physical activity [27], and low cardiorespiratory fitness [28]. Additionally, weight status may also impact upon psychological outcomes–such as self-esteem, anxiety, and depression [29]–which in turn influence academic achievement. Alternatively, it is possible that, in some cultural contexts at least, the association between academic achievement and weight status is complex and affected by localised experiences, such as ethnicity or racialised identity, socioeconomic status, and urbanicity.

As a contribution to this literature, we examined the association between weight status and academic achievement in New Caledonia, a French island located in the South Pacific. Overweight and obesity affect a large proportion of New Caledonian adolescents, although there are marked ethnic differences [3]. Specifically, the population of New Caledonia is multi-ethnic, with part of the population being of Melanesian origin (45%, locally called "Kanak"), while individuals of European and Polynesian origin represent 31% and 12% of the population respectively [30]. These different ethnic groups have different lifestyles: most Kanak have retained a traditional tribal lifestyle, while Europeans are typically more Westernised [3, 31]. Moreover, while obesity disproportionately affects Oceanian, Non-European, Non-Asian (ONENA) adolescents in New Caledonia compared to those of European origin [3], the outcomes of weight status may also differ across ethnic groups. For example, there is some evidence that higher BMIs are associated with lower self-esteem and greater body dissatisfaction in adolescents of European but not ONENA origin [17].

Importantly, there are also ethnic differences in academic achievement both in the Pacific region generally and in New Caledonia specifically. For example, in 2014, 36% of Kanaks had no diploma, compared with 17% of non-Kanaks. Only 5% of Kanaks had obtained a higher education diploma, compared to 28% of non-Kanaks [32]. In 2009, non-Kanaks were almost ten times more likely to obtain a higher education diploma than Kanaks [33]. The intersection between ethnicity and gender may also be important in New Caledonia, with girls typically doing better than boys at school, with a higher graduation rate [33]. Additionally, the New Caledonian context is marked by large urban-rural differences, especially when compared with schooling in mainland France or the French Overseas Territories. Among the 30 colleges whose remoteness index were greater than ten across the entire French territory, 15 are in New Caledonia [34]. Furthermore, in New Caledonia, the most isolated schools typically welcome more socially disadvantaged students [34] and combine practically all *a priori* factors unfavourable to educational success, which explains the poorer academic achievement of students in more rural areas [34].

Kanak adolescents in New Caledonia, therefore, may be more likely than adolescents of European origin to experience many of the unfavourable factors that are detrimental to academic achievement, including higher weight status, lower SES, and attending school in more rural areas [3]. To date, however, no previous research has specifically examined the extent to which weight status is associated with academic achievement in this multicultural context, particularly in terms of ethnic differences. In the present study, therefore, we sought to overcome this gap in the literature by examining associations between academic achievement and weight status in New Caledonian adolescents of Kanak and European origin. Additionally, we also examined whether any putative relationship remained robust once covariates had been accounted for, namely socioeconomic status, gender identity, area of residence (urban vs. rural), and school remoteness.

## Materials and methods

### Participants and procedures

In this cross-sectional study, the sample included 526 New Caledonian adolescents (257 boys, 269 girls). The research fulfilled all legal obligations and followed the principles of the Declaration of Helsinki. The protocol was approved by the Ethics Committee of New Caledonia (CCE 2018–06 001), and all parents provided written informed consent before their adolescents participated in the study. Anthropometric data were collected between July 2018 and April 2019 and academic achievement tests were completed in December 2018, 2019, and 2020 (see below).

To determine the appropriate sample size for the present research, a power analysis was undertaken through G*Power 3.1.9.7. In absence of previous work in Pacific region, we estimated the effect size to be around 0.15, based on the effect size ($r$ = .164) indicated by He et al. in their meta-analysis [23]. For a 4 x 2 analysis of variance ($df$ = 3) with $\alpha$ = .05 and power of .80, we required a minimum sample size of $N$ = 500, which was surpassed in our study.

New Caledonia comprises of three provinces (North Province, South Province, and Loyalty Islands Province), with notable difference in terms of ethnic distribution, SES, and urbanisation. Eight secondary public schools were selected: one in the Loyalty Islands Province (greater rurality), two in the North Province (east and west coasts, rural areas), and five in the South Province, three of which were in the capital, Noumea (the only urbanise area in New Caledonia). This selection yielded a representative selection of students between rural and urban area (63% and 37%, respectively) and between the different provinces (70% in South Province, 20% in North Province, and 10% in Loyalty Islands Province). Our selection criterion included school size ($N$ > 200) to ensure sufficient data in a single field trip. Based on this criterion,

only one school was eligible in the Loyalty Islands Province, five schools were eligible in the North Province, and nineteen in the South Province (eight in Noumea). Schools were then randomly selected and contacted to obtain staff agreement. A staff member then randomly selected six grades (6th, 7th, 8th, or 9th year), for a total of approximately 150 students (six groups with a mean of 25 students per grade). All students in each class were asked to participate in the study without exclusion criteria. We obtained 90% of possible participant data. Non-participation ($n = 124$) occurred for a variety of reasons, including a small number of parents declining to take part, others failing to return parental informed consent forms, or the absence of participants. Adolescents with missing anthropometric data ($n = 74$) were excluded from the analyses (S1 Fig).

### Measures

**Academic achievement.** Academic achievement was assessed through results of the test passed by all the French students at the end of the ninth grade [35], each year in December in New Caledonia. Data were obtained for each participant from the Statistical Department of the Vice Rectorate of New Caledonia. Scores were produced in four academic domains: French literature and language (100 points), mathematics (100 points), history, geography, and moral and civic education (50 points), and physics, chemistry, life and earth sciences, and technology (any two of the three; 50 points). We summed the scores to obtain a global academic achievement score, ranging from 0 to 300. Participants took these tests in 2018, 2019, or 2020 depending on their school grade during the data collection process (7th, 8th, or 9th year in 2018 or 2019). Due to the COVID-19 pandemic, no data were obtained in 2021 because testing was canceled ($n = 272$). Moreover, we did not have academic achievement scores for 52 students for various reasons: absent during the exam ($n = 12$), not found in the database, probably because they left New Caledonia ($n = 29$), or because they had taken a professional exam ($n = 11$). An additional 32 students were excluded because they took their exam in 2022, after the COVID-19 pandemic (S1 Fig).

**Anthropometric measurements.** All anthropometric data were collected by trained member of the research team in the school nursing office. Height was measured to the nearest 0.1 cm using a portable stadiometer (Leicester Tanita HR 001, Tanita Corporation, Tokyo, Japan). Weight was recorded to the nearest 0.1 kg with a scale (Tanita HA 503, Tanita Corporation) while adolescents wore light clothing. The body mass index (BMI) was subsequently calculated by dividing the weight (in kilograms) by the height (in meters) squared.

Body mass index (BMI) was then calculated by dividing weight (in kilograms) by height (in meters) squared. For weight status classifications, three BMI-based definitions were used: the International Obesity Task Force's BMI-based references [36], the World Health Organization's BMI-based references [37], and French BMI-based references [38]. BMI z-scores (BMI-z) were calculated using the LMS reference values for each BMI-based classification.

Waist circumference was measured at the midpoint between the lowest rib and the iliac crest. The measuring tape was placed horizontally and measurements were done twice at the end of a normal expiration, to the nearest 0.1 cm. If the difference between the two recorded measurements was greater than 0.5 cm, a third measurement was taken, and the mean of the two nearest values was recorded. Waist-to-height ratio (WHtR = waist circumference in cm divided by height in cm) was then calculated. Categorization of WHtR was done using the National Institute for Health and Care Excellence (NICE) guidelines to assess and predict health risks: 'no increased risk' (WHtR < 0.5), 'increased risk' (WHtR = 0.5 to < 0.6), and 'very high risk' (WHtR > 0.6) [39].

The thickness of triceps, biceps, supra-iliac, and subscapular skinfolds was measured twice to the nearest 0.1 mm using Holtain skinfold calipers. Next, the sum of the four skinfolds (S4SF) was calculated as a proxy for body fatness.

**Ethnicity.**    Ethnicity was self-reported by the adolescents and categorized according to the INSERM report on New Caledonia [40], with the modification that the participants could only select one ethnic group. Participants from Polynesian ($n = 68$) and other ethnic ($n = 38$) groups were excluded from our analysis due to the small subsample sizes (S1 Fig).

**Sociodemographic data.**    Socioeconomic status (SES) and area of residence were determined as previously described [3]. SES was based on the occupation of the household reference person (the householder with the highest income) and categorized in three groups: high SES (managerial and professional occupations), intermediate SES (intermediate occupations), and low SES (routine and manual occupations). Area of residence was determined using a European standard [41].

**School remoteness index (SRI).**    The "remoteness" index makes it is possible to summarize the notion of remoteness for a given establishment as a continuous indicator [42, 43]. This index takes into 17 variables distributed into three types: variables relating to the students, variables relating to educational opportunities present around the establishment, and variables relating to sporting and cultural opportunities present around the establishment [42]. The SRI was obtain from each school from the statistical department of the Vice Rectorate of New Caledonia [44]. Higher SRI scores reflect greater school remoteness.

## Statistical analysis

All analyses were conducted using IBM SPSS v.25. We first examined between-group (Kanak vs. European) differences in sample characteristics using univariate analyses of variance (ANOVAs) or chi-squared tests. Pearson's bivariate correlations were used to assess the relationship between academic achievement and weight status indicators in each ethnic group. To further interrogate between-group differences, we computed a 4 x 2 ANOVA with ethnicity (Kanak vs. European) and weight status (underweight, normal weight, overweight and obese) entered as independent variables and academic achievement scores as the dependent variable. Next, hierarchical linear analysis was used to examine whether the association between academic achievement and weight status remained stable after controlling for covariates. Academic achievement scores were entered as the criterion variable and the following variables were entered in a first step: gender identity, SES, and SRI. Gender and SES were categorized into groups by creating dummy variables. The year of taking the exam (2018, 2019, or 2020) was also entered in the regression in the first step to account for the variability of the exam each year. BMI-$z$ (used as a proxy for all weight status indicators) was entered in a second step. However, because WHtR is also found to have significant relationship with academic achievement, we also computed sensitivity analysis using this anthropometric parameter.

## Results

### Sample characteristics

Table 1 presents descriptive statistics of the study sample as a function of ethnicity. Preliminary analyses indicated significant between-group differences in SES (European adolescents had significantly higher SES than Kanak adolescents), the distribution of gender identity (there were more boys in the European subsample and more girls in Kanak subsample), and the distribution of area of residence (Kanak adolescents mostly lived in rural areas, while European mostly lived in urban areas). There were no significant between-group differences in the distribution of the year in which the exam was taken. There were also significant between-group

**Table 1. Sociodemographic characteristics as a function of ethnicity.**

| | | Kanak (n = 278) | | European (n = 248) | | | |
|---|---|---|---|---|---|---|---|
| | | **n** | **%** | **n** | **%** | **p** | **Cramer's V** |
| Gender identity | Boys | 119 | 42.8 | 138 | 55.6 | .003 | .128 |
| | Girls | 159 | 57.2 | 110 | 44.4 | | |
| Socioeconomic status | High | 39 | 14.0 | 137 | 55.2 | < .001 | .494 |
| | Intermediate | 29 | 10.4 | 43 | 17.3 | | |
| | Low | 210 | 75.5 | 68 | 27.4 | | |
| Year of exam | 2018 | 65 | 23.4 | 60 | 24.2 | .181 | .081 |
| | 2019 | 137 | 49.3 | 104 | 41.9 | | |
| | 2020 | 76 | 27.3 | 84 | 33.9 | | |
| Weigh Status (using IOTF ref.) | underweight | 10 | 3.6 | 28 | 11.3 | < .001 | .268 |
| | Normal | 162 | 58.3 | 180 | 72.6 | | |
| | Overweight | 76 | 27.3 | 29 | 11.7 | | |
| | Obesity | 30 | 10.8 | 11 | 4.4 | | |
| | | **M** | **SD** | **M** | **SD** | **p** | **Cohen's d** |
| School remoteness index | | 10.32 | 10.86 | 2.67 | 9.69 | < .001 | 0.74 |
| BMI z-score (IOTF ref.) | | .87 | 1.06 | .22 | 1.12 | < .001 | 0.59 |
| BMI z-score (WHO ref.) | | .77 | 1.18 | .07 | 1.23 | < .001 | 0.58 |
| BMI z-score (French ref.) | | 1.20 | 1.38 | .39 | 1.38 | < .001 | 0.58 |
| WHtR | | .49 | .07 | .46 | .06 | < .001 | 0.46 |
| S4SF | | 58.1 | 36.1 | 48.1 | 31.2 | .001 | 0.30 |
| Academic achievement score 2018 | | 165.1 | 49.5 | 157.3 | 49.9 | .383 | 0.16 |
| Academic achievement score 2019 | | 143.1 | 41.7 | 206.0 | 47.1 | < .001 | 1.41 |
| Academic achievement score 2020 | | 136.2 | 40.1 | 182.6 | 55.4 | < .001 | 0.96 |

differences in SRI (European adolescents had significantly lower scores than adolescents from Kanak origin) and for all anthropometric indicators of adiposity, with Kanak adolescents having a higher BMI z-scores, WHtR, and S4SF than European adolescents. Finally, there were significant between-group differences in academic achievement scores, with European adolescents having higher academic achievement than Kanak adolescents in 2019 and in 2020 but not in 2018.

## Association between academic achievement results and obesity indicator

Pearson's bivariate correlations were computed between academic achievement, weight status indices (i.e., BMI-z, WHtR, and S4SF; see Table 2) for each ethnic group separately. Negative

**Table 2. Associations between indicators of weight status and academic achievement in European (top diagonal) and Kanak (bottom diagonal) adolescents.**

| | **(1)** | **(2)** | **(3)** | **(4)** | **(5)** | **(6)** |
|---|---|---|---|---|---|---|
| (1) Academic achievement | | -.190* | -.244* | -.202* | -.203* | -.209* |
| (2) S4SF | -.032 | | .861* | .737* | .753* | .777* |
| (3) WHtR | -.066 | .852* | | .810* | .825* | .839* |
| (4) BMI-z (IOTF ref.) | -.051 | .775* | .854* | | .997* | .996* |
| (5) BMI-z (WHO ref.) | -.050 | .779* | .862* | .998* | | .995* |
| (6) BMI-z (French ref.) | -.047 | .798* | .875* | .996* | .998* | |

*p < .001. S4SF = Sum of the four skinfolds; WHtR = Waist-to-height ratio; BMI = Body mass index.

weak-to-moderate correlations were found between academic achievement and all anthropometric parameters in European adolescents, but the same associations did not reach significance in Kanak adolescents. For all further analyses, we used BMI-$z$ (IOT ref.) as a proxy for weight status because it is widely used to determine the weight status of adolescents in other international studies. However, because WHtR was found to have a higher relationship with academic achievement, we also computed sensitivity analysis using this anthropometric parameter.

## Weight status and ethnic differences in academic achievement

To further identify between-group differences in academic achievement as a function of weight status, we computed a 4 x 2 ANOVA with weight status and ethnicity entered as independent variables and academic achievement scores as the dependent variable. The results showed a significant interaction between ethnicity and weight status, $F(3, 526) = 3.07$, $p = .028$, $\eta_p^2 = .02$. There was also a significant main effect of ethnicity, $F(1, 526) = 15.03$, $p < .001$, $\eta_p^2 = .03$. Finally, there was a significant main effect of weight status, $F(3, 525) = 4.99$, $p = .002$, $\eta_p^2 = .03$ (see Fig 1). When comparing academic achievement between European and Kanak adolescents, we found a significant difference for underweight ($p = .037$) and normal weight ($p < .001$) adolescents (Fig 1). For overweight and obese adolescents, there was no significant difference in academic achievement between European and Kanak adolescents ($p = .098$ and $p = .875$, respectively).

Similar results were obtained in a sensitivity analysis using WHtR instead of BMI-$z$ as an indicator of weight status (S2 Fig), with European adolescents with WHtRs < 0.5 and those with WHtRs = 0.5 to < 0.6 having significantly higher academic achievement than Kanak adolescents in the same WHtR categories ($p < .001$ in each case).

## Hierarchical linear analyses

Because there appeared to be ethnic differences in the relationship between weight status and academic achievement in the correlational analyses, we conducted hierarchical linear analyses with each ethnic group separately (see Table 3). In adolescents of European origin, the second step of the regression was significant, with higher academic achievement significantly

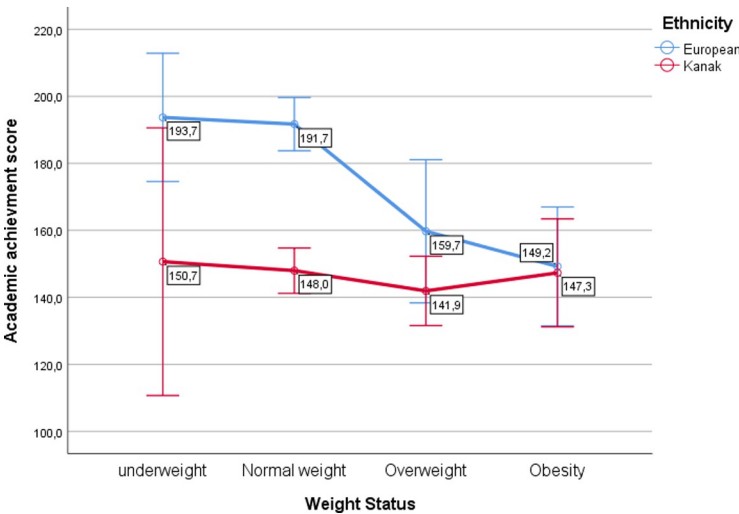

**Fig 1. Academic achievement as a function of weight status and ethnicity.**

**Table 3. Hierarchical linear analysis models examining predictors of academic achievement in European and Kanak adolescents.**

| | European (n = 248) | | | | | Kanak (n = 278) | | | | |
|---|---|---|---|---|---|---|---|---|---|---|
| **Step 1** | | | | | | | | | | |
| $R^2$ (Adj. $R^2$) | .25 (.23) | | | | | .12 (.10) | | | | |
| *ΔF* | 13.61 ($p < .001$) | | | | | 5.89 ($p < .001$) | | | | |
| **Variable** | *B* | *SE* | *β* | *t* | *p* | *B* | *SE* | *β* | *t* | *p* |
| SRI | **-1.24** | **.33** | **-.222** | **-3.75** | **< .001** | -.07 | .24 | -.018 | -.30 | .761 |
| Girls† | 7.69 | 6.27 | .071 | 1.23 | .221 | **13.07** | **5.18** | **.146** | **2.53** | **.012** |
| High SES£ | **27.98** | **7.29** | **.258** | **3.84** | **< .001** | **22.98** | **7.42** | **.180** | **3.10** | **.002** |
| Mid SES£ | 17.86 | 9.44 | .125 | 1.89 | .060 | 16.27 | 8.52 | .112 | 1.91 | .057 |
| 2019 exam# | **52.52** | **7.84** | **.480** | **6.70** | **< .001** | **-22.48** | **6.36** | **-.254** | **-3.53** | **< .001** |
| 2020 exam# | **30.11** | **8.18** | **.254** | **3.68** | **< .001** | **-28.76** | **7.19** | **.289** | **-4.00** | **< .001** |
| **Step 2** | | | | | | | | | | |
| $R^2$ (Adj. $R^2$) | .27 (.25) | | | | | .12 (.10) | | | | |
| *ΔF* | 4.50 ($p = .035$) | | | | | 1.07 ($p = .302$) | | | | |
| | *B* | SE | *β* | *t* | *p* | *B* | SE | *β* | *t* | *p* |
| BMI-*z* | **-5.91** | **2.79** | **-.122** | **-2.12** | **.035** | -2.51 | 2.43 | -.060 | -1.03 | .302 |
| SRI | **-1.20** | **.33** | **-.215** | **-3.67** | **< .001** | -.08 | .24 | -.021 | -.35 | .724 |
| Girls† | 6.50 | 6.25 | .060 | 1.04 | .299 | **13.56** | **5.20** | **.151** | **2.61** | **.010** |
| High SES£ | **24.48** | **7.42** | **.226** | **3.30** | **.001** | **22.46** | **7.43** | **.176** | **3.02** | **.003** |
| Mid SES£ | 13.78 | 9.56 | .097 | 1.44 | .151 | 16.01 | 8.53 | .110 | 1.88 | .061 |
| 2019 exam# | **51.70** | **7.79** | **.473** | **6.63** | **< .001** | **-23.20** | **6.40** | **-.262** | **-3.63** | **< .001** |
| 2020 exam# | **29.25** | **8.13** | **.256** | **3.60** | **< .001** | **-28.72** | **7.19** | **-.289** | **-3.99** | **< .001** |

SRI: School remoteness index; SES, socioeconomic status. Bold text indicates a statistically significant difference with a *p*-value less than .05. Reference is: † Boys; £ Low SES, #2018 exam

associated with lower BMI-*z*, lower school remoteness, and higher SES. In Kanak adolescents, the second step of the regression did not reach significance. In the first step of the regression, greater academic achievement was significantly associated with higher SES and gender identity (girls had higher academic achievement than boys). Similar results were obtained in a sensitivity analysis using WHtR instead of BMI-*z* as an indicator of weight status (S1 Table).

## Discussion

In the present study, we examined the relationship between academic achievement and weight status indicators in a sample of school-age adolescents from New Caledonia. Our findings showed, firstly, that all weight status indicators (i.e., BMI-*z*, WHtR, and S4SF) were significantly and inversely associated with academic achievement in adolescents of European origin. In contrast, the same associations did not reach significance in Kanak adolescents. Between-group comparisons showed that the largest ethnic differences in academic achievement occurred in adolescents who were in the "underweight and normal weight" category. Additionally, linear model analysis showed that the association between BMI-*z* and academic achievements remained significant in adolescents of European origin after controlling for the effects of gender identity, SES, school remoteness, and exam year. In contrast, the second step of the regression, which included BMI-*z*, was not significant in Kanak adolescents. Overall, these results suggest that higher weight status is associated with lower academic achievement, but only in New Caledonian adolescents of European origin.

What might explain these different effects for each ethnic group? One commonly utilized explanation is that higher weight status directly affects cognition function, compromising attention and set-shifting [45], inhibitory control [46], and abstract reasoning [47]. However, if this were the case, we should expect similar effects across both ethnic groups included in the present study. An alternative possibility is that the relationship between weight status and academic achievement is differentially impacted by one's ethno-cultural background. In this view, it is the meaning and lived experience of higher weight status within particular cultural and ethnic groups that may drive any impact upon academic achievement. For instance, to the extent that weight status is more strongly or closely related to personhood and perceptions of self-identity, its impact upon outcomes in various life domains can be expected to be stronger [48, 49]. Conversely, weaker or even null associations between weight status and outcomes in those life domains can be expected in communities where the association between body size and self-schemas are largely decoupled.

More specifically, it is possible that the sociocultural importance of thinness and slender body sizes is greater in New Caledonian adolescents of European origin. For this ethnic group in particular, therefore, personhood and identity may be more strongly coupled with lower weight status [17]. If this is the case, then we might also expect that higher weight status in adolescents of European origin impacts psychological outcomes, such as self-esteem [17], which in turn detrimentally affect academic achievement. In contrast, Kanak adolescents may be more likely to decouple weight status from self-schemas than adolescents of European origin, or indeed may positively perceive a wide range of body sizes as ideal for both historic and contemporaneous reasons [48]. In this case, weight status may have an attenuated or null effect on potential variables that mediate or moderate any impact on academic achievement, such as self-esteem [17].

In short, therefore, the present results suggest that the association between weight status and academic achievement may not be straightforward, at least in the New Caledonian context. It is likely that the way that weight status is experienced and perceived–both individually and collectively–shapes any relationship with outcomes such as academic achievement. In broad outline, this conclusion is consistent with the view that there are cultural differences in the strength of the association between weight status and academic achievement [23, 24]. In contrast to previous work that has focused on cultural differences at the level of nations or geographic regions, however, the present work suggests that ethno-cultural experiences within nations may also shape the ways that weight status impacts upon academic achievement. This corroborates some previous work showing that the association between weight status and academic achievement is not significant in Black children from the United States [50].

An alternative explanation for the present findings is that the null relationship between weight status and academic achievement in Kanak adolescents reflects the fact that other factors may be more important than weight status in this group. For instance, our linear analyses showed that both gender identity and SES were significantly associated with academic achievement in Kanak adolescents, which is consistent with previous work [51–53]. Thus, it may be that–for Kanak adolescents, at least–gender identity and SES may be more important in shaping academic outcomes, with weight status being less relevant. Conversely, in adolescents of European origin, our results showed that weight status continued to matter in terms of academic achievement even after accounting for the significant effects of school remoteness and SES. That is, for adolescents of European origin, weight status appears to remain a meaningful contributor to academic achievement beyond the influence of significant covariates.

A strength of the present work lies in the focus on a population that remains relatively neglected within the broader literature. Indeed, as identified previously [23], there is a paucity of research from the region generally. Nevertheless, a number of limitations of the present

work should be considered. First, because our study design was cross-sectional, we are unable to draw causal conclusions. This is important because it is possible that there are complex bi-directional links between weight status and academic achievement, at least for our adolescents of European origin. Second, because of the COVID-19 pandemic, no data were available from 2021 because testing was canceled. As a result, some participants had to be excluded from analyses. Relatedly, given the method of recruitment, our participants were probably not representative of the broader population of New Caledonian adolescents. Third, our analyses were focused on objective indicators of weight status, but there is some evidence that self-perceived weight status may be more strongly associated with academic achievement [54, 55]. This may have been particular important given evidence that self-perceived weight status may differ across ethnic groups in New Caledonia [56].

## Conclusion

These limitations notwithstanding, the present study makes an important contribution to the literature examining associations between weight status and academic achievement. Beyond shining a light on a community that has been historically neglected in the academic literature, the results of the present study also suggest that the impact of weight status on academic achievement may vary as a function of ethno-cultural background. This is important because it highlights the need for effective school-based interventions aimed to alleviating any potential impact of obesity on academic achievement. For instance, promoting healthy and safe school environments to help students develop effective coping skills to deal with issues related to weight may be helpful, particularly for students of European descent in New Caledonia. Additionally, while routine monitoring and management of health conditions related to obesity remains important, there may be additional support needs for overweight adolescents (e.g., behavioural counselling, social support). In terms of theory, there is now a need to more fully understand the reasons why weight status differentially related to academic achievement, a task that may be suited to qualitative research methods. Doing so may help scholars to identify factors that may heighten the impact of weight status on academic achievement for some students, while also determining whether there may be some experiences that help to limit or attenuate the impact of weight status on academic achievement.

## Supporting information

**S1 Fig. Flow diagram showing participants assessed for eligibility.**
(TIF)

**S2 Fig. Academic achievement as a function of WHtR and ethnicity.**
(TIF)

**S1 Table. Hierarchical linear regression models examining predictors of academic achievement in European and Kanak adolescents.**
(DOCX)

## Acknowledgments

We would like to thank the school teaching teams and administrative staffs for their help and support in our investigations, especially the Vice-Rectorat of New Caledonia. We would like to thank Christophe Serra-Mallol, Emilie Paufique, Paul Zongo, Solange Ponidja, Pierre-Yves Le Roux, and Fabrice Wacalie for their help for collecting the data. The funding sources had no role in study design, data collection, data analysis, interpretation and writing of the report.

## Author Contributions

**Conceptualization:** Stéphane Frayon, Olivier Galy.

**Data curation:** Guillaume Wattelez.

**Formal analysis:** Stéphane Frayon.

**Funding acquisition:** Stéphane Frayon, Olivier Galy.

**Investigation:** Guillaume Wattelez, Akila Nedjar-Guerre, Olivier Galy.

**Methodology:** Stéphane Frayon, Olivier Galy.

**Project administration:** Olivier Galy.

**Writing – original draft:** Stéphane Frayon, Viren Swami.

**Writing – review & editing:** Viren Swami, Guillaume Wattelez, Akila Nedjar-Guerre, Olivier Galy.

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
