## [Decision Letter · Decision Letter 0]

18 Jun 2024

PONE-D-24-14640Associations between academic achievement and weight status in a multi-ethnic sample of New Caledonian adolescentsPLOS ONE

Dear Dr. Frayon,

Thank you for submitting your manuscript to PLOS ONE. After careful consideration, we feel that it has merit but does not fully meet PLOS ONE’s publication criteria as it currently stands. Therefore, we invite you to submit a revised version of the manuscript that addresses the points raised during the review process.

Please submit your revised manuscript by Aug 02 2024 11:59PM. If you will need more time than this to complete your revisions, please reply to this message or contact the journal office at plosone@plos.org. Please include the following items when submitting your revised manuscript:A rebuttal letter that responds to each point raised by the academic editor and reviewer(s). You should upload this letter as a separate file labeled 'Response to Reviewers'.A marked-up copy of your manuscript that highlights changes made to the original version. You should upload this as a separate file labeled 'Revised Manuscript with Track Changes'.An unmarked version of your revised paper without tracked changes. You should upload this as a separate file labeled 'Manuscript'.If applicable, we recommend that you deposit your laboratory protocols in protocols.io to enhance the reproducibility of your results. Protocols.io assigns your protocol its own identifier (DOI) so that it can be cited independently in the future. For instructions see: https://journals.plos.org/plosone/s/submission-guidelines#loc-laboratory-protocols. Additionally, PLOS ONE offers an option for publishing peer-reviewed Lab Protocol articles, which describe protocols hosted on protocols.io. Read more information on sharing protocols at https://plos.org/protocols?utm_medium=editorial-email&utm_source=authorletters&utm_campaign=protocols.

We look forward to receiving your revised manuscript.

Kind regards,

Deepak Dhamnetiya, MD

Academic Editor

PLOS ONE

Journal Requirements:

-https://doi.org/10.1186/s40359-022-01032-y 

In your revision ensure you cite all your sources (including your own works), and quote or rephrase any duplicated text outside the methods section. Further consideration is dependent on these concerns being addressed.

"This project was funded by the University of New Caledonia and the Fondation Nestlé France. "

Reviewers' comments:

Reviewer's Responses to Questions

**Comments to the Author**

1. Is the manuscript technically sound, and do the data support the conclusions?

Reviewer #1: Yes

Reviewer #2: Yes

2. Has the statistical analysis been performed appropriately and rigorously? 

Reviewer #1: Yes

Reviewer #2: Yes

3. Have the authors made all data underlying the findings in their manuscript fully available?

Reviewer #1: Yes

Reviewer #2: Yes

4. Is the manuscript presented in an intelligible fashion and written in standard English?

Reviewer #1: Yes

Reviewer #2: Yes

5. Review Comments to the Author

Reviewer #1: I would like to appreciate authors for selecting such an interesting topic addressing the vulnerable adolescent age group.

I would like to provide few suggestions.

1)Abstract:

In methodology kindly mention the study duration.

2) Introduction:

The authors first can start with global burden of obesity and current trend with increase in obesity among adolescents.

To restrict introduction to 3-4 paragraphs.

3) Methodology:

Kindly mention the study design. More detail description of participant selection have to be provided like inclusion criteria, total number of secondary schools and why only 8 eight schools were selected. What are exactly divisions.

The authors have mentioned

that study period commenced from 2018-2019, I would like to know why authors are commenting regarding 2020 and 2021 year. As it is understood the data collection period does not include these years.

The authors have to mention who were measuring the anthropometric measurements in research team, are they are trained for measuring required anthropometric measurements?.

4) Conclusion:

Kindly include concluding points on European and Kanak ethnic groups.

Reviewer #2: Dear Author, The paper has significant findings.However, some clarifications are needed

Comments:

in the introduction, what is the nutritional status of students in New Caledonia

the sample size calculation was not stated

study locations: how many schools are in urban and rural?

Table 3 should be in scientific table

How do you imply the results into practice?

6. PLOS authors have the option to publish the peer review history of their article (what does this mean?). If published, this will include your full peer review and any attached files.

Reviewer #1: No

Reviewer #2: No

---

## [Author Response · Author response to Decision Letter 0]

2 Jul 2024

We thank the reviewers and editors for their appraisals. We have done our best to address the concerns and hope that the manuscript is now suitable for publication. Please note that all the change has been tracked in red. 

Journal Requirements:

This has now been done, as requested. 

We checked and rewrote the sentences concerned

3. Please state what role the funders took in the study. 

This has been done. 

4. Please review your reference list to ensure that it is complete and correct.

This has been verified. 

Reviewer #1: 

1)Abstract:

In methodology kindly mention the study duration.

This has now been done, as requested. 

2) Introduction:

The authors first can start with global burden of obesity and current trend with increase in obesity among adolescents.

To restrict introduction to 3-4 paragraphs.

Thank you for this suggestion. We have revised the opening paragraph to include some information about the global burden of obesity, as requested. However, we have struggled to comply with the second request, namely to substantive shorten the Introduction. In our view, all the information currently presented in the Introduction is necessary to provide a full and rounded review of the literature, as well as to provide a clear justification for the present work. As such, while we have made minor edits to shorten the Introduction, we have not made any substantive revisions in this regard. 

3) Methodology:

Kindly mention the study design. 

This has been done, please see line 107. 

More detail description of participant selection have to be provided like inclusion criteria, total number of secondary schools and why only 8 eight schools were selected.

All the inclusion criteria and the rational to selected eight schools has been described in the manuscript. See lines 123-134. 

What are exactly divisions.

We apologise for any confusion. We now use the term “grade” throughout, which should hopefully be more familiar to readers. 

The authors have mentioned that study period commenced from 2018-2019, I would like to know why authors are commenting regarding 2020 and 2021 year. As it is understood the data collection period does not include these years.

We apologise for any lack of clarity. As specified in the manuscript lines 151-152, participants took academic tests at the end of the ninth grade in 2018, 2019, or 2020 depending on their school grade during the data collection process (7th, 8th, or 9th year in 2018 or 2019).

The authors have to mention who were measuring the anthropometric measurements in research team, are they are trained for measuring required anthropometric measurements?

This has been done Please line 160. 

4) Conclusion:

Kindly include concluding points on European and Kanak ethnic groups.

Thank you for this suggestion. In the final paragraph of the Discussion, we now briefly discussion implications of our findings. 

Reviewer #2: Dear Author, The paper has significant findings.However, some clarifications are needed

Comments:

1. in the introduction, what is the nutritional status of students in New Caledonia

We apologise, but it is unclear what further information the reviewer is requesting. If the reviewer is requesting information about nutritional intake vs. nutritional demands of adolescents in New Caledonia, we are sorry to say that such data are not currently available. 

2. the sample size calculation was not stated

This has been done Please line 117-122. 

3. study locations: how many schools are in urban and rural?

This has been specified, please see line 123-129. 

4. Table 3 should be in scientific table

We apologise that this was not done earlier. All tables have now been formatted according to the journal’s guidelines. 

5. How do you imply the results into practice?

Thank you for this suggestion. In the final paragraph of the Discussion, we now briefly discussion implications of our findings.

---

## [Decision Letter · Decision Letter 1]

20 Aug 2024

Associations between academic achievement and weight status in a multi-ethnic sample of New Caledonian adolescents

PONE-D-24-14640R1

Dear Dr. Frayon,

We’re pleased to inform you that your manuscript has been judged scientifically suitable for publication and will be formally accepted for publication once it meets all outstanding technical requirements.

Kind regards,

Deepak Dhamnetiya, MD

Academic Editor

PLOS ONE

Additional Editor Comments (optional):

Reviewers' comments:

Reviewer's Responses to Questions

**Comments to the Author**

1. If the authors have adequately addressed your comments raised in a previous round of review and you feel that this manuscript is now acceptable for publication, you may indicate that here to bypass the “Comments to the Author” section, enter your conflict of interest statement in the “Confidential to Editor” section, and submit your "Accept" recommendation.

Reviewer #3: All comments have been addressed

Reviewer #4: All comments have been addressed

Reviewer #5: (No Response)

2. Is the manuscript technically sound, and do the data support the conclusions?

Reviewer #3: Yes

Reviewer #4: Yes

Reviewer #5: Partly

3. Has the statistical analysis been performed appropriately and rigorously? 

Reviewer #3: Yes

Reviewer #4: Yes

Reviewer #5: Yes

4. Have the authors made all data underlying the findings in their manuscript fully available?

Reviewer #3: Yes

Reviewer #4: Yes

Reviewer #5: Yes

5. Is the manuscript presented in an intelligible fashion and written in standard English?

Reviewer #3: Yes

Reviewer #4: Yes

Reviewer #5: Yes

6. Review Comments to the Author

Reviewer #3: Thank you for making changes as requested by the reviewers. The manuscript is in good shape now. Hope the paper gains audience so that similar studies could be carried out in larger scale, with a more robust and primary dataset.

Reviewer #4: Author were able to make a substantial improvement from the previous version of the article. There are minor typographical errors that needs to be adjusted like in page 3, line 30.

The article can be published

Reviewer #5: I will like to thank the authors and the editor for the opportunity to review and learn rom this piece of work.

This a good addition to knowledge on adolescent academic achievement; which has great implications for educational policy and practice. It also bring to the fore neglected populations and their academic struggles.

However, some observations have been made in the abstract, methodology, result and references. These observations have been highlighted on the manuscript reviewed which has been uploaded as "Reviewers attachment".

Thank you, once, again.

7. PLOS authors have the option to publish the peer review history of their article (what does this mean?). If published, this will include your full peer review and any attached files.

Reviewer #3: **Yes: **Aftab Ahmad

Reviewer #4: **Yes: **Abdulrahman Ahmad

Reviewer #5: **Yes: **Philip Adewale Adeoye (Jos, Nigeria)

---

## [Editor Report · Acceptance letter]

23 Sep 2024

PONE-D-24-14640R1 

PLOS ONE

Dear Dr. Frayon, 

I'm pleased to inform you that your manuscript has been deemed suitable for publication in PLOS ONE. Congratulations! Your manuscript is now being handed over to our production team.

Kind regards, 

on behalf of

Dr. Deepak Dhamnetiya 

Academic Editor

PLOS ONE